# Learning Multi-Agent Communication with Contrastive Learning

**Yat Long Lo**
Dyson Robot Learning Lab
richie.lo@dyson.com

**Biswa Sengupta**
Imperial College London
biswasengupta@gmail.com

**Jakob Foerster**
FLAIR, University of Oxford
jakob.foerster@eng.ox.ac.uk

**Michael Noukhovitch**
Mila
Université de Montréal
mnoukhov@gmail.com

## Abstract

Communication is a powerful tool for coordination in multi-agent RL. But inducing an effective, common language is a difficult challenge, particularly in the decentralized setting. In this work, we introduce an alternative perspective where communicative messages sent between agents are considered as different incomplete views of the environment state. By examining the relationship between messages sent and received, we propose to learn to communicate using contrastive learning to maximize the mutual information between messages of a given trajectory. In communication-essential environments, our method outperforms previous work in both performance and learning speed. Using qualitative metrics and representation probing, we show that our method induces more symmetric communication and captures global state information from the environment. Overall, we show the power of contrastive learning and the importance of leveraging messages as encodings for effective communication.

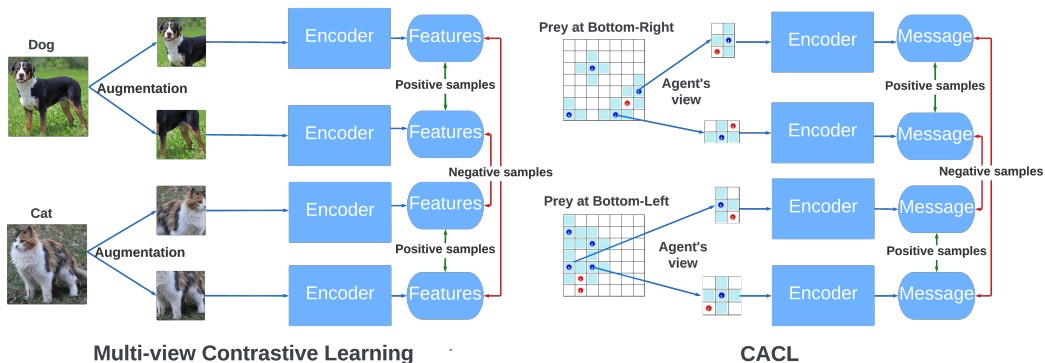

Figure 1: In multi-view learning, augmentations of the original image or "views" are used as positive samples for contrastive learning. In our proposed method, CACL, different agents' views of the same environment state are considered positive samples and messages are contrastively learned as encodings of that state.

# 1 Introduction

Communication is a key capability necessary for effective coordination among agents in partially observable environments. In multi-agent reinforcement learning (MARL) (Sutton & Barto, 2018), agents can use their actions to transmit information (Grupen et al., 2020) but continuous or discrete

messages on a communication channel (Foerster et al., 2016), i.e., linguistic communication (Lazaridou & Baroni, 2020), are more flexible and powerful because they can convey more complex concepts. To successfully communicate, a speaker and a listener must share a common language with a shared understanding of the symbols being used (Skyrms, 2010; Dafoe et al., 2020). Emergent communication or learning a common protocol (Wagner et al., 2003; Lazaridou & Baroni, 2020), is a thriving research direction but most works focus on simple, single-turn, sender-receiver games (Lazaridou et al., 2018; Chaabouni et al., 2019). In more visually and structurally complex MARL environments (Samvelyan et al., 2019), existing approaches often rely on centralized learning mechanisms by sharing models (Lowe et al., 2017) or gradients (Sukhbaatar et al., 2016).

However, a centralized controller is impractical in many real-world environments (Mai et al., 2021; Jung et al., 2021) where agents cannot easily synchronize and must act independently i.e. decentralized. Centralized training with decentralized execution (CTDE) (Lowe et al., 2017) is a middle-ground between purely centralized and decentralized methods but may not perform better than purely decentralized training (Lyu et al., 2021). A centralized controller suffers from the *curse of dimensionality*: as the number of agents it must control increases, the amount of communication between agents to process increases exponentially (Jin et al., 2021). Furthermore, the fully decentralized setting is more flexible and requires fewer assumptions about other agents, making it more realistic in many real-world scenarios (Li et al., 2020). Hence, this work explores learning to communicate to coordinate agents in the decentralized setting. In MARL, this means each agent will have its own model to decide how to act and communicate, and no agents share parameters or gradients.

Typical RL approaches to decentralized communication are known to perform poorly even in simple tasks (Foerster et al., 2016) due to the large space of communication to explore, the high variance of RL, and a lack of common grounding on which to base communication (Lin et al., 2021). Earlier work leveraged how communication influences other agents (Jaques et al., 2018; Eccles et al., 2019) to learn the protocol. Most recently, Lin et al. (2021) proposed agents that autoencode their observations and use the encodings as communication, using the shared environment as the common grounding. We build on this work in using both the shared environment and the relationship between sent and received messages to ground a protocol. We extend the Lin et al. (2021) perspective that agents' messages are encodings and propose that agents in similar states should produce similar messages. This perspective leads to a simple method based on contrastive learning to ground communication.

Inspired by the literature in representation learning that uses different "views" of a data sample (Bachman et al., 2019), for a given trajectory, we frame an agent's observation as a "view" of the environment state. Thus, different agents' messages are encodings of different incomplete "views" of the same underlying state. From this perspective, messages from the same state should, generally, be more similar to each other than to those from distant states or other trajectories, as shown in Figure 1. As with image augmentations, two agent observations may not necessarily overlap, but a contrastive objective can generally lead to effective encodings. We propose Communication Alignment Contrastive Learning (CACL), where each agent separately uses contrastive learning between their own sent and received messages to learn a communication encoding.

We experimentally validate CACL in three communication-essential environments and show how CACL leads to improved performance and speed, outperforming state-of-the-art decentralized MARL communication algorithms. To understand CACL's success, we propose a suite of qualitative and quantitative metrics. We demonstrate that CACL leads to more symmetric communication (i.e., different agents communicate similarly when faced with the same observations), allowing agents to be more mutually intelligible. By treating our messages as representations, we show that CACL's messages capture global semantic information about the environment better than baselines. Overall, we argue that contrastive learning is a powerful direction for multi-agent communication and has fundamental benefits over previous approaches.

## 2  RELATED WORK

Learning to coordinate multiple RL agents is a challenging and unsolved task where naively applying single-agent RL algorithms often fails (Foerster et al., 2016). Recent approaches focus on neural network-based agents (Goodfellow et al., 2016) with a message channel to develop a common communication protocol (Lazaridou & Baroni, 2020). To handle issues of non-stationarity, some work focuses on centralized learning approaches that globally share models (Foerster et al., 2016),

training procedures (Lowe et al., 2017), or gradients (Sukhbaatar et al., 2016) among agents. This improves coordination and can reduce optimization issues but results are often still sub-optimal in practise (Foerster et al., 2016; Lin et al., 2021) and may violate independence assumptions, effectively modelling the multi-agent scenario as a single agent (Eccles et al., 2019).

This work focuses on independent, decentralized agents and non-differentiable communication. In previous work, Jaques et al. (2018) propose a loss to influence other agents but require explicit and complex models of other agents and their experiments focus on mixed cooperative-competitive scenarios. Eccles et al. (2019) add biases to each agent's loss function that separately encourage positive listening (i.e., the listener to act differently for different messages) and positive signaling (i.e., the speaker to produce diverse messages in different situations). Their method is simpler but requires task-specific hyperparameter tuning to achieve reasonable performance and underperforms in sensory-rich environments (Lin et al., 2021). Our work is closest to Lin et al. (2021), who leverage autoencoding as their method to learn a message protocol in cooperative 2D MARL games. Agents learn to reconstruct their observations and communicate their autoencoding. It outperforms previous works while being algorithmically and conceptually simpler. Our method builds on this encoding perspective by considering other agents' messages to ground communication. Whereas agents in Lin et al. (2021) can only learn to encode the observation, our approach leverages the relationship between different agents' messages to encode global state information. Empirically, our method is also more efficient as it requires no extra learning parameters whereas Lin et al. (2021) learn and discard their decoder network. Note that our setup uses continuous messages instead of discrete (Eccles et al., 2019; Lin et al., 2021), a standard choice in contrastive learning (Chopra et al., 2005; He et al., 2020; Chen et al., 2020a) and embodied multi-agent communication (Sukhbaatar et al., 2016; Jiang & Lu, 2018; Das et al., 2019).

Autoencoding is a form of generative self-supervised learning (SSL) (Doersch et al., 2015). We propose to use another form of SSL, contrastive learning (Chen et al., 2020a), as the basis for learning communication. We are motivated by recent work that achieves state-of-the-art representation learning on images using contrastive learning methods (Chen et al., 2020b) and leverages multiple "views" of the data. Whereas negative samples are simply different images, positive samples are image data augmentations or "views" of the original image (Bachman et al., 2019). We treat agents' messages of the same state in a trajectory as positives of each other, so we base our method on SupCon (Supervised Contrastive Learning) (Khosla et al., 2020) which modifies the classic contrastive objective to account for multiple positive samples. Relatedly, Dessì et al. (2021) use a two-agent discrete communication setup to do contrastive learning on images, we do the opposite and leverage contrastive learning to learn multi-agent communication in an RL environment.

## 3 PRELIMINARIES

We base our investigations on decentralized partially observable Markov decision processes (Dec-POMDPs) with $N$ agents to describe a *fully cooperative multi-agent task* (Oliehoek & Amato, 2016). A Dec-POMDP consists of a tuple $G = \langle S, A, P, R, Z, \Omega, n, \gamma \rangle$. $s \in S$ is the true state of the environment. At each time step, each agent $i \in N$ chooses an action $a^i \in A^i$ to form a joint action $a \in A \equiv A^1 \times A^2... \times A^N$. It leads to an environment transition according to the transition function $P(s'|s, a^1, ...a^N) : S \times A \times S \rightarrow [0, 1]$. All agents share the same reward function $R(s, a) : S \times A \rightarrow \mathbb{R}$. $\gamma \in [0, 1)$ is a discount factor. As the environment is partially observable, each agent $i$ receives individual observations $z \in Z$ based on the observation function $\Omega^i(s) : S \rightarrow Z$.

We denote the environment trajectory and the action-observation history (AOH) of an agent $i$ as $\tau_t = s_0, a_0, ....s_t, a_t$ and $\tau_t^i = \Omega^i(s_0), a_0^i, ....\Omega^i(s_t), a_t^i \in T \equiv (Z \times A)^*$ respectively. A stochastic policy $\pi(a^i|\tau^i) : T \times A \rightarrow [0, 1]$ conditions on AOH. The joint policy $\pi$ has a corresponding action-value function $Q^\pi(s_t, a_t) = \mathbb{E}_{s_{t+1:\infty}, a_{t+1:\infty}}[R_t|s_t, a_t]$, where $R_t = \sum_{i=0}^{\infty} \gamma^i r_{t+i}$ is the discounted return. $r_{t+i}$ is the reward obtained at time $t + i$ from the reward function $R$.

To account for communication, similar to Lin et al. (2021), at each time step $t$, an agent $i$ takes an action $a_t^i$ and produces a message $m_t^i = \Psi^i(\Omega^i(s_t))$ after receiving its observation $\Omega^i(s_t)$ including messages from the previous time step $m_{t-1}^{-i}$, where $\Psi^i$ is agent $i$'s function to produce a message given its observation and $m_{t-1}^{-i}$ refers to messages sent by agents other than agent $i$. The messages are continuous vectors of dimensionality $D$.

## 4    METHODOLOGY

We propose a different perspective on the message space used for communication. At each time step $t$ for a given trajectory $\tau$, a message $m_t^i$ of an agent $i$ can be viewed as an incomplete view of the environment state $s_t$ because $m_t^i$ is a function of $s_t$ as formulated in section 3. Naturally, messages of all the $N$ agents are different incomplete perspectives of $s_t$. To ground decentralized communication, we hypothesize that we could leverage this relationship between messages from similar states to encourage consistency and similarity of the messages space across agents. Specifically, we propose maximizing the mutual information using contrastive learning which aligns the message space by pushing messages from similar states closer together and messages of different states further apart. Note that agents see a partial view of the state from their observation, so they will inherently communicate different messages to reflect their partial knowledge. However, aligning their message spaces enables communicating these partial views of the state in a more mutually-intelligible way.

As a heuristic for state similarity, we consider a window of timesteps within a trajectory to be all similar states i.e. positive samples of each other. To guarantee dissimilar negative samples (Schroff et al., 2015), we use states from other trajectories as negatives. Since each underlying state has multiple positive views ($w$ steps, $N$ agent messages each), we leverage the recent contrastive learning method SupCon (Khosla et al., 2020). We refer to the contrastive SupCon objective across multiple MARL trajectories as *Communication Alignment Contrastive Learning (CACL)*.

Let $M$ be all the messages in a batch of trajectories and $M_\tau$ be the messages in trajectory $\tau$. Let $m_t^i \in M_\tau$ the message of agent $i$ at time $t$. Thus, positives $H$ for a message $m_t^i$ given a timestep window $w$ are all other messages from the same trajectory $\tau$ sent within that timestep window $H(m_t^i) \equiv \{m_{t'}^j \in M_\tau \setminus \{m_t^i\} : t' \in [t-w, t+w]\}$. Let all other messages $K$ from all trajectories in the batch be $K(m_t^i) \equiv M \setminus \{m_t^i\}$. Formally, the contrastive loss $L_{CACL}$:

$$\sum_{m_t^i \in M_\tau} \frac{-1}{|H(m_t^i)|} \sum_{m_h \in H(m_t^i)} \log \frac{\exp(m_t^i \cdot m_h / \eta)}{\sum_{m_k \in K(m_t^i)} \exp(m_t^i \cdot m_k / \eta)} \tag{1}$$

Where $\eta \in \mathbb{R}^+$ is a scalar temperature and $|H(m_t^i)|$ is the cardinality. Practically, each agent has a replay buffer that maintains a batch of trajectory data collected from multiple environment instances. It contains messages received during training to compute the *CACL* loss. We use a timestep window of size 5 for all the environments based on hyperparameter tuning of different window sizes. Following Khosla et al. (2020), messages are normalized before the loss computation and a low temperature (i.e. $\eta = 0.1$) is used as it empirically benefits performance and training stability. The total loss for each agent is a reinforcement learning loss $L_{RL}$ using the reward to learn a policy (but not message head) and a separate contrastive loss $L_{CACL}$ to learn just the message head, formulated as follows:

$$L = L_{RL} + \kappa L_{CACL} \tag{2}$$

where $\kappa \in \mathbb{R}^+$ is a hyperparameter to scale the *CACL* loss. See Appendix A.1 for $L_{RL}$ details.

## 5    EXPERIMENTS AND RESULTS

### 5.1    EXPERIMENTAL SETUP

We evaluate our method on three multi-agent environments with communication channels. Given the limited information each agent observes, agents must meaningfully communicate in order to improve task performance. All results are averaged over 12 evaluation episodes and over 6 random seeds. More details of the environments and parameters can be found in appendix A.2.

**Traffic-Junction**: Proposed by Sukhbaatar et al. (2016), it consists of a 4-way traffic junction with cars entering and leaving the grid. The goal is to avoid collision when crossing the junction. We use 5 agents with a vision of 1. Though unnecessary, with limited vision in agents, communication could help to solve the task. We evaluate algorithms using their success rate in avoiding all collisions during evaluation episodes.

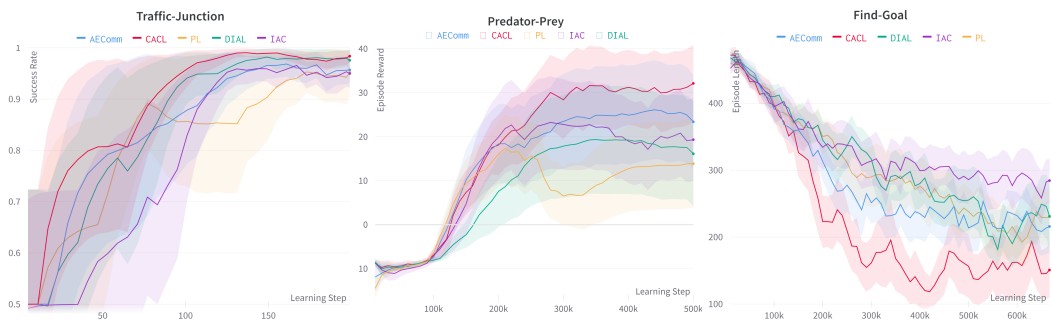

Figure 2: CACL (red) outperforms all other methods on Traffic-Junction (left), Predator-Prey (left) and Find-Goal (right). Predator-Prey shows evaluation reward, higher is better. Traffic-Junction plots the percent of successful episodes, higher is better. Find-Goal plots the episode length until the goal is reached, lower is better. The performance curves are smoothed by a factor of 0.5 with standard errors plotted as shaded areas.

**Predator-Prey**: A variant of the classic game (Benda et al., 1986; Barrett et al., 2011) based on Koul (2019) where 4 agents (i.e. predators) have the cooperative goal to capture 2 randomly-moving prey by surrounding each prey with more than one predator. We devise a more difficult variation where agents have to entirely surround a prey on all 4 sides to successfully capture it and they cannot see each other in their observations. Thus, agents must communicate their positions and actions in order to coordinate their attacks. We evaluate each algorithm with episodic rewards in evaluation episodes.

**Find-Goal**: Proposed by Lin et al. (2021), agents' goal is to reach the green goal location as fast as possible in a grid environment with obstacles. We use 3 agents, each observes a partial view of the environment centered at its current position. Unlike in Lin et al. (2021), we use a field of view of $3 \times 3$ instead of $7 \times 7$ to make the problem harder. Each agent receives an individual reward of 1 for reaching the goal and an additional reward of 5 when all of them reach the goal. Hence, it is beneficial for an agent to communicate the goal location once it observes the goal. As in Lin et al. (2021), we measure performance using episode length. An episode ends quicker if agents can communicate goal locations to each other more efficiently. Hence, a method performs better if it has shorter episode lengths.

## 5.2 TRAINING DETAILS

We compare CACL to the state-of-the-art independent, decentralized method, autoencoded communication (AEComm; Lin et al., 2021), which grounds communication by reconstructing encoded observations. We also compare to baselines from previous work: independent actor critic without communication (IAC) and positive listening loss (PL; Eccles et al., 2019) (See Appendix A.6). We exclude the positive signalling loss (Eccles et al., 2019) as extending it to continuous messages is non-trivial but note that AEComm outperforms it in the discrete case (Lin et al., 2021). We also include DIAL (Foerster et al., 2016) which learns to communicate through differentiable messages to share gradients so is decentralized but not independent.

All methods use the same architecture based on the IAC algorithm with n-step returns and asynchronous environments (Mnih et al., 2016). Each agent has an encoder for observations and received messages. For methods with communication, each agent has a communication head to produce messages based on encoded observations. For policy learning, a GRU (Cho et al., 2014) is used to generate a hidden representation from a history of observations and messages. Agents use the hidden state for their the policy and value heads, which are 3-layer fully-connected neural networks. We perform spectral normalization (Gogianu et al., 2021) in the penultimate layer for each head to improve training stability. The architecture is shown in Figure 8 and hyperparameters are further described, both in Appendix A.3.

## 5.3 TASK PERFORMANCE

We run all methods on the three selected environments and plot results in Figure 2. Our proposed method CACL outperforms all baseline methods in both final performance and learning speed and,

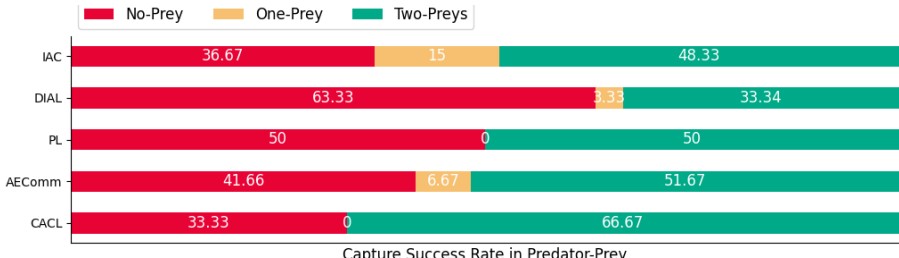

Figure 3: Success rate in Predator-Prey: the percentage of final evaluation runs that captured no prey, one prey, or both prey. Average over 6 random seeds, each with 10 evaluation episodes. See Appendix 3 for the same results with standard deviation.

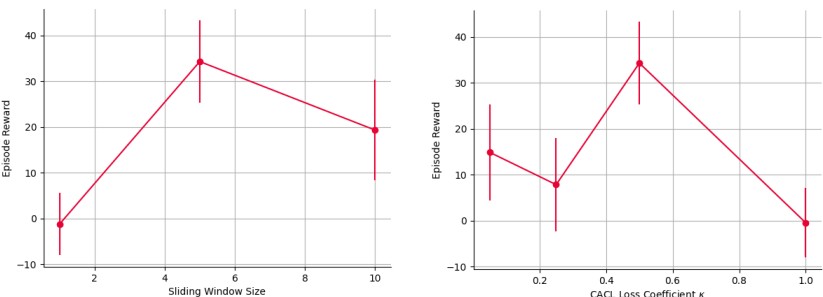

Figure 4: Predator-Prey ablation experiment on $L_{CACL}$ varying the sliding window size and $\kappa$.

consistent with previous results (Lin et al., 2021), AEComm is the strongest baseline. The largest performance increase from CACL is in FindGoal where partial observability is most prominent due to agents' small field-of-view which makes communication more necessary (hence why IAC performs worst). These results show the effectiveness of self-supervised methods for learning communication in the fully-decentralized setting, as they both outperform DIAL which, notably, backpropogates gradients through other agents. Furthermore, it demonstrates CACL's contrastive learning as a more powerful alternative to AEComm's autoencoding for coordinating agents with communication.

Improvement on Traffic-Junction is not as significant as others because communication is less essential for task completion, as shown by the strong performance of IAC. For Predator-Prey, results are clearly better than baselines but have high variance due to the difficulty of the task. The goal of Predatory-Prey is to capture two moving prey and requires coordinating precisely to surround and attack a prey at the same time. Any slight miscoordination leads to sharp drop in rewards. For another metric of success, we compute the percentages of evaluation episodes that capture no, one, or two preys. Averaging over 6 random seeds, we show results in Figure 3. CACL does significantly better on the task, outperforms all baselines, and solving the complete task more robustly while failing less frequently. Find-Goal requires the most communication among the environments because the gridworld is the largest and agents must clearly communicate the location of goal. Here, CACL significantly outperforms the baselines, demonstrating that as the communicative task gets harder, CACL outperforms more.

We confirm the effectiveness of CACL with an ablation study of the key design decisions: sliding window and SupCon. CACL leverages the temporal nature of RL to treat a sliding window of timesteps as positive views of each other. We plot results for a range of window sizes run on Predator-Prey in Figure 4. No sliding window (size 1) performs poorly, demonstrating its necessity and that the choice of sliding window size is an important hyperparameter. Through the use of SupCon (Khosla et al., 2020) we treat all sent and received messages in the sliding window as all positive views of each other, with many positives per batch. Creating a batch with just one positive view per message corresponds to SimCLR (Chen et al., 2020a) and results in much worse performance ($1.36 \pm 9.46$). We also run Predator-Prey and search across values of the CACL loss coefficient $\kappa$ in Figure 4. We used the best values (5-step window, $\kappa$=0.5) across all the environments, demonstrating that the choice of CACL hyperparameters is robust. Overall, we show the issues in naively implementing contrastive learning for communication, and the clear, important design decisions behind CACL.

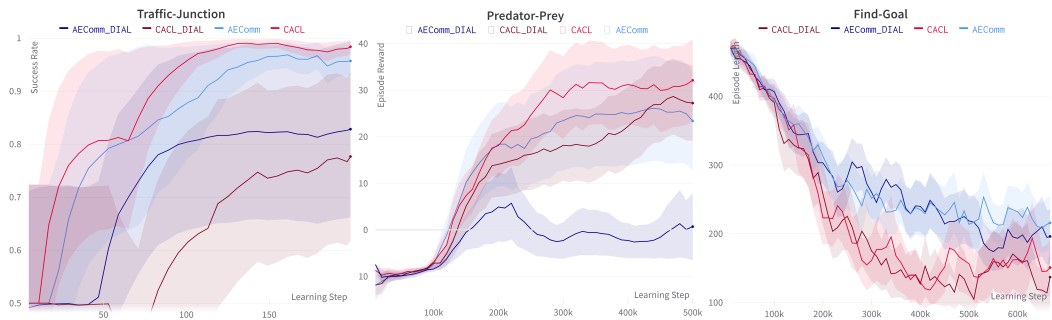

Figure 5: Comparing CACL and AEComm with their respective variants when combined with DIAL. Variants with DIAL have generally worse performance.

## 5.4 AUGMENTING CACL WITH RL

The contrastive loss in the communication head of CACL is very performant without optimizing for reward, so a natural question is whether we can achieve even better results if we learn the message using reward as well. To answer this, we add DIAL to both CACL and the next best method, AEComm, and evaluate in the three environments. This is equivalent to backpropogating $L_{RL}$ from Equation 2 through agents to learn the message head. In this way, both RL and SSL (contrastive or autoencoding) signals are used to learn the message head.

Figure 5 compares the performance of CACL and AEComm with their DIAL-augmented variants. Our findings are consistent with Lin et al. (2021), who find that mixing AEComm and RL objectives are detrimental to performance. We observe that augmenting either AEComm or CACL with DIAL performs generally worse, except in Find-Goal, where performances is similar but not better. We hypothesize that decentralized DIAL is a complex, and high-variance optimization that is difficult to stabilize. DIAL's gradient updates may clash with CACL and result in neither a useful contrastive representation, nor a strong reward-oriented one. It is also possible that CACL's messages would not be improved with reward-oriented gradients. As we show in Section 5.6, CACL already captures useful semantic information that other agents can effectively extract.

## 5.5 PROTOCOL SYMMETRY

We hypothesize that CACL's improved performance over the baselines is because it induces a more consistent, mutually-intelligible communication protocol that is shared among agents. More specifically, we consider consistency to be how similarly agents communicate (i.e., sending similar messages) when faced with the same observations. A consistent protocol can reduce the optimization complexity since agents only need to learn one protocol for the whole group and it also makes agents more mutually intelligible.

Table 1: Protocol symmetry across environments, average and standard deviation over 10 episodes and 6 random seeds. CACL consistently learns the most symmetric protocol.

|  | DIAL | PL | AEComm | CACL (Ours) |
|---|---|---|---|---|
| Predator-Prey | $0.66 \pm 0.07$ | $0.66 \pm 0.06$ | $0.89 \pm 0.01$ | $\mathbf{0.95 \pm 0.01}$ |
| FindGoal | $0.50 \pm 0.05$ | $0.49 \pm 0.04$ | $0.85 \pm 0.02$ | $\mathbf{0.92 \pm 0.01}$ |
| Traffic Junction | $0.69 \pm 0.01$ | $0.61 \pm 0.04$ | $0.80 \pm 0.01$ | $\mathbf{0.98 \pm 0.002}$ |

To evaluate consistency, we measure protocol symmetry (Graesser et al., 2019) so if an agent swaps observations and trajectory with another agent, it should produce a similar message as what the other agent produced. We extend this metric from previous work to the continuous, embodied case. We feed the same trajectory to all agents and measure the pairwise cosine similarities of the messages that they produce. Given a trajectory $\tau$ and $\{t \in T\}$ as a set of time steps of $\tau$, protocol symmetry ($protocol\_sym$) is written as:

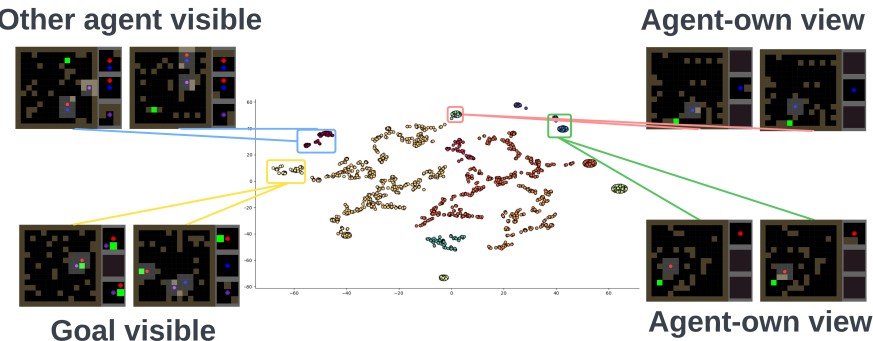

Figure 6: DBSCAN (Ester et al., 1996) clustering results of messages produced by CACL after dimensionality reduction with t-SNE. We show four semantically meaningful clusters with their corresponding labels: messages sent when the goal is visible, when another agent is visible, and two clusters correspond to only the individual agents visible in the observation.

$$\frac{1}{|T|}\sum_{t \in T}\frac{1}{|N|}\sum_{i \in N}\frac{1}{|N|-1}\sum_{j \in N \setminus i}\frac{\Psi^i(\Omega^j(s_t)) \cdot \Psi^j(\Omega^j(s_t))}{\|\Psi^i(\Omega^j(s_t))\|\|\Psi^j(\Omega^j(s_t))\|} \qquad (3)$$

Therefore, a more consistent protocol has higher symmetry. We swap agent trajectory and observations and compute this metric over 10 evaluation episodes for 6 random seeds, and show results in Table 1. The self-supervised methods (CACL and AEComm) clearly outperform the others (DIAL and PL) implying that SSL is better for learning consistent representations in decentralized MARL. Furthermore, CACL's protocol is very highly symmetric, clearly outperforming all others. Each AEComm agent autoencodes their own observation without considering the other agents' messages, leading to the formation of multiple protocols between agents. In contrast, CACL induces a common protocol by casting the problem in the multi-view perspective and implicitly aligning agents' messages. The possible correlation between protocol symmetry and overall performance and speed further indicates the benefits of learning a common protocol in the decentralized setting.

## 5.6    PROTOCOL REPRESENTATION PROBING

To further investigate how informative our protocols are, we propose a suite of qualitative and quantitative representation probing tests based on message clustering and classification, respectively. We perform these tests on the protocols learned in the Find-Goal environment.

Similar to Lin et al. (2021), we cluster messages generated from 10 evaluation episodes to qualitatively assess how informative CACL's protocol is. The messages are first compressed to a dimension of 2 using t-SNE (Van der Maaten & Hinton, 2008) and then clustered using DBSCAN (Ester et al., 1996). We look at each cluster's messages and their corresponding observations to extract any patterns and semantics captured. As shown in Figure 6, we observe a cluster of messages for observations when the goal is visible and another one when another agent is visible. Two clusters correspond to agents seeing neither the goal nor another agent. Notably, the messages in these clusters can come from different agents in different episodes, demonstrating that agents can indeed communicate symmetrically. The clusters indicate that CACL learns to compress meaningful, global state information in messages, allowing other agents to reasonably learn this semantic information.  To quantitatively evaluate the

Table 2: Classification results of the two probing tests in Find-Goal. All methods perform similarly in the easier Goal Visibility Test while CACL outperforms the baselines significantly in the more difficult Goal Location Test.

|  | DIAL | PL | AEComm | CACL (Ours) |
|---|---|---|---|---|
| Goal Visibility | $99.45\% \pm 2.68$ | $98.87\% \pm 0.67$ | $99.75\% \pm 0.04$ | $97.75\% \pm 0.69$ |
| Goal Location | $68.15\% \pm 1.76$ | $78.31\% \pm 2.39$ | $76.14\% \pm 3.36$ | $\mathbf{91.28\% \pm 1.71}$ |

informativeness of learned protocols, we propose to treat messages as representations and learn a

classifier on top of the messages, following work in RL representation learning (Lazaridou et al., 2018; Anand et al., 2019). Since FindGoal is focused on reaching a goal, intuitively, agents should communicate whether they have found the goal and, if so, where other agents should go to reach the goal. Thus, we propose to probe the goal visibility and goal location. The former uses the messages to classify whether the goal is visible in observations or not (i.e. a binary classification). The latter uses messages where the goal is visible in the observations to classify the general location of the goal (i.e., a 5-class classification: Top-Left, Top-Right, Bottom-Left, Bottom-Right and Middle). Whereas goal visibility is easy for egocentric communication, goal location requires detailed spatial information and communicating the absolute location from their relative position. This tests whether the communication protocol can consider other agents' perspectives and give global information from an egocentric observation. We use 30 evaluation episodes per method to generate messages for our experiments but different methods may have different numbers of acceptable messages for our probing task (e.g. a limited number of messages where the goal is visible for predicting goal location). To ensure fair comparison, we choose an equal number of samples per class (i.e., positive/negative, 5-class location) for all methods and use a 70%/30% random split for training and testing. We use a 2-layer fully-connected neural network to test each method, as this corresponds to the same network that agents use to encode each others' messages as part of their observations.

Table 2 shows the classification results for the two probing tests. Goal visibility is an easier task and all methods' messages can be effectively used to determine it. In the more difficult goal location task, all methods perform above chance (20%) but CACL's protocol significantly outperforms baselines. Contrastive learning across different agents' messages can enable CACL to learn a more global understanding of location from their egocentric viewpoint. We further compare the similarity between messages sent when the goal is at location $[1, 1]$ and when the goal is at other coordinates along the diagonal in Appendix A.7, which corroborates with the Goal Location Test in CACL's capability in encoding global information better. By encoding the goal's spatial information, CACL agents are more likely able to move directly towards it, and reduce episode length. If other methods simply communicate that a goal is found, agents know to alter their search but are not as precise in direction. This explains why AEComm, PL, and DIAL perform better than IAC but worse than CACL, which also learns much quicker as shown in Figure 2. For completeness, we also provide similar classification results with a one-layer (linear) probe in Appendix A.8.

## 6    Limitations

Our work investigates fully-cooperative environments but learning to communicate in less cooperative settings, such as those with adversaries (Noukhovitch et al., 2021), is a harder optimization problem. CACL would likely need stronger regularization to be effective. Furthermore, our empirical testing has revealed that SSL objectives are ineffective with reward-oriented gradients, as demonstrated in section 5.4. Although this phenomenon is well known (Lin et al., 2021), it is still not fully understood and future work should aim to combine the two objectives. Finally, this work evaluates agents that were trained together. A more challenging frontier is zero-shot communication, an extension of zero-shot cooperation (Hu et al., 2020), in which agents must communicate effectively with novel partners, unseen during training. In Appendix A.9, we show how existing methods perform poorly in this settings and leave this challenging setup to future work.

## 7    Conclusion and Future Work

This work introduces an alternative perspective for learning to communicate in decentralized MARL based on the relationship between sent and received messages within a trajectory. Drawing inspiration from multi-view learning, we ground communication using contrastive learning by considering agents' messages to be encoded views of the same state. We empirically show that our method leads to better performance through a more consistent, common protocol and learns to communicate more global state information. We believe this work solidifies contrastive learning as an effective perspective for learning to communicate and hope it invigorates research into contrastive methods for communication with a focus on consistency. Furthermore, by establishing the connection between multi-view SSL, which has traditionally focused on images, and communication in MARL, we hope to encourage more cross-domain research. Finally, we see contrastive learning as a potential method for simulating human language evolution, and hope to inspire research in this direction.

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

# A  APPENDIX

## A.1  REINFORCEMENT LEARNING LOSS

In Equation 2, we use $L_{RL}$ to denote the RL loss. As our investigation is orthogonal to the choice of RL algorithm, this term can be any RL algorithm dependent on which algorithm is being used. In this work, we use Independent Asynchronous Actor Critic (IA2C) as our base algorithm, denoted as IAC in our experiments (Christianos et al., 2020). Given an agent $i \in N$, it has a policy $\pi_\phi^i$ and value function $V_\theta^i$, parameterized by parameters $\phi$ and $\theta$ respectively. The policy loss for agent $i$ is defined as:

$$L(\phi) = -\log(\pi(a_t^i|\Omega^i(s_t); \phi)(r_t + \gamma V(\Omega^i(s_{t+1}); \theta) - V(\Omega^i(s_t); \theta)) \tag{4}$$

with the value function minimizes:

$$L(\theta) = \|V(\Omega^i(s_t); \theta) - (r_t + \gamma V(\Omega^i(s_{t+1}); \theta))\|^2 \tag{5}$$

These two losses are denoted as $L_{RL}$ due to space limit. In the case with communication, the local observation of an agent is simply augmented with received messages.

## A.2  ENVIRONMENT DETAILS

Figure A.2 provides a visual illustration of the environments used.

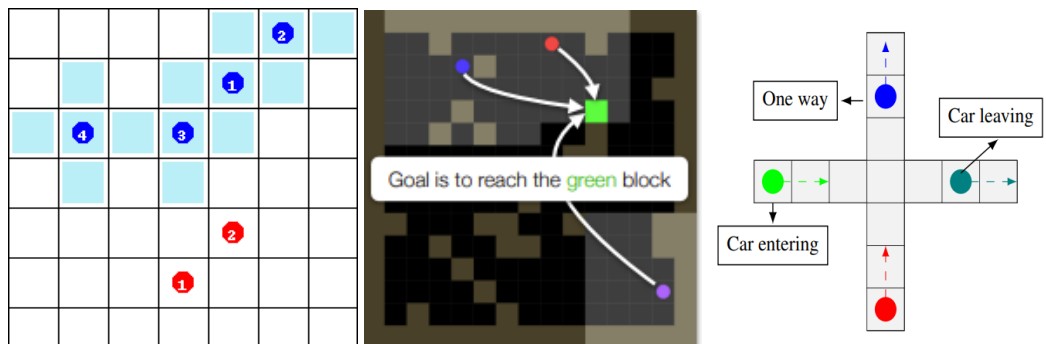

Figure 7: Visual illustration of the environments used. Left: Predator-Prey, taken from Koul (2019). Middle: Find-Goal, taken from Lin et al. (2021). Right: Traffic-Junction, taken from Singh et al. (2018)

## A.2.1  PREDATOR-PREY

We modify the Predator-Prey implementation by Koul (2019). Our Predator-Prey has a higher communication and coordination requirement than the original Predator-Prey environment. Specifically, for a prey to be captured, it has to be entirely surrounded (i.e. the prey cannot move to another grid position in any actions).

Here, we use an 7x7 gridworld. In each agent's observation, it can only see the prey if it is within the field of view (3x3) and cannot see where other agents are. A shared reward of 10 is given for a successful capture and a penalty of -0.5 is given for a failed attempt. A -0.01 step penalty is also applied per step. Each agent has the actions of *LEFT*, *RIGHT*, *UP*. *DOWN* and *NO-OP*. The prey has the movement probability vector of $[0.175, 0.175, 0.175, 0.175, 0.3]$ with each value corresponding to the probability of each action taken.

All algorithms are trained for 30 million environment steps with a maximum of 200 steps per episode.

### A.2.2 FIND-GOAL

We use the Find-Goal environment implementation provided by Lin et al. (2021). The agents have the goal to find where the goal is in a 15x15 grid world with obstacles.

Unlike in Lin et al. (2021), each agent has a 3x3 field of view (instead of 7x7) to make the task more difficult. Each agent receives a reward of 1 for reaching a goal and an additional reward of 5 if all agents reach the goal. We use a step penalty of -0.01 and an obstacle density of 0.15.

All algorithms are trained for 40 million environment steps with a maximum of 512 steps per episode.

### A.2.3 TRAFFIC-JUNCTION

We use the Traffic-Junction environment implementation provided by Singh et al. (2018). The gridworld is 7x7 with 1 traffic junction. The rate of cars being added has a minimum and maximum of 0.1 and 0.3. We use the easy version with two arrival points and 5 agents. Agents are heavily penalized if a collision happens and have only two actions, namely *gas* and *brake*.

All algorithms are trained for 20 million environment steps with a maximum of 20 steps per episode.

### A.3 ARCHITECTURE AND HYPERPARAMETERS

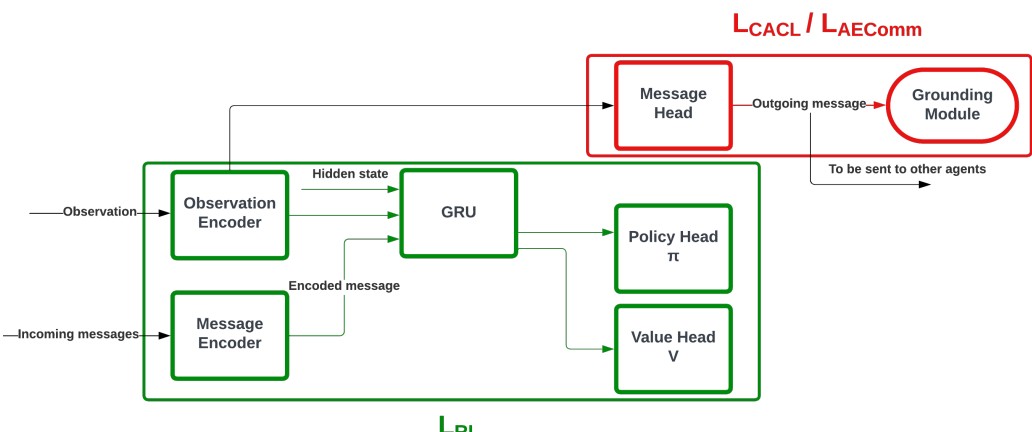

Figure 8: Architectural illustration for algorithms with communication. To remove communication, the message head is disabled. Grounding module is only relevant to CACL and AEComm. The former is a loss function and the latter is a decoder to reconstruct the encoded observation. Green boxes denote components where gradients of $L_{RL}$ are applied to to. Red boxes denote components where gradients of $L_{CACL}$ and $L_{AEComm}$ are applied to. $L_{AEComm}$ refers to the loss function of AEComm (Lin et al., 2021).

Figure 8 illustrates the components of the architecture used in this work, similar to (Lin et al., 2021). A message head is only used for algorithms with communication, namely CACL, AEComm, PL and DIAL. The Grounding Module refers to mechanisms to ground the messages produced by the message head, used in CACL and AEComm. Colored boxes denote components where gradients are applied to for a particular loss function. In the case of DIAL, gradients would flow from another agent to the agent that sends a message through the message head and message encoder. Unless specified otherwise, we fix all hidden layers to be a size of 32.

We experimented with using the output of the GRU, or hidden state, to condition the message head. Empirically we found that directly conditioning on the observation encoding, as in Lin et al. (2021), led to more stable learning dynamics.

The observation encoder output values of size 32. For Predator-Prey and Traffic Junction, a one-layer fully-connected neural network is used as observation encoder. For Find-Goal, same as Lin et al.

Table 3: Success rate in Predator-Prey: the percentage of final evaluation runs that captured no prey, one prey, or both prey. Average and standard deviation over 6 random seeds.

|  | No-Prey | One-Prey | Two-Preys |
|---|---|---|---|
| IAC | $36.67\% \pm 7.50$ | $15.00\% \pm 3.57$ | $48.33\% \pm 6.83$ |
| DIAL | $63.33\% \pm 7.56$ | $3.33\% \pm 1.24$ | $33.34\% \pm 7.86$ |
| PL | $50.00\% \pm 8.33$ | $0.00\% \pm 0.00$ | $50.00\% \pm 8.63$ |
| AEComm | $41.66\% \pm 7.48$ | $6.67\% \pm 1.84$ | $51.67\% \pm 7.60$ |
| CACL (Ours) | $\mathbf{33.33\% \pm 7.86}$ | $0.00\% \pm 0.00$ | $\mathbf{66.67\% \pm 7.86}$ |

(2021), we use a two-layer convolutional neural network followed by a 3-layer fully-connected neural network.

For the message encoder, it outputs values of size 8 in Predator-Prey and Find-Goal with one hidden layer. It outputs values of size 16 in Traffic-Junction with two hidden layers. These configurations are selected based on the best performance of the baseline communication learning algorithm used - DIAL. Messages received are concatenated before passing to message encoders. For all the methods with communication, they produce messages of length 4 ($D = 4$) with a sigmoid function as activation. All models are trained with the Adam optimizer (Kingma & Ba, 2014).

Table 4 lists out the hyperparameters used for all the methods.

### A.4 CACL CONTRASTIVE LOSS CURVE

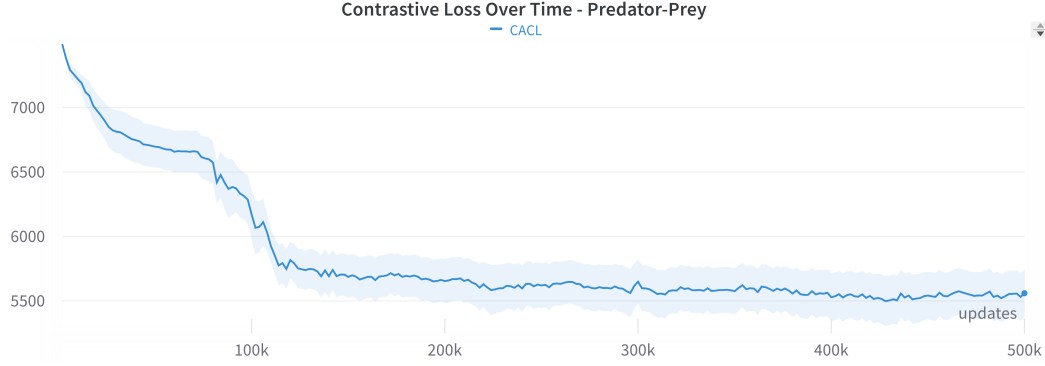

Figure 9: CACL's contrastive loss over time during training for Predator-Prey

Figure 9 shows the CACL's contrastive loss over time during training for Predator-Prey. This illustrates training convergence of the loss and improved separation of positive and negative samples between the start and end of training.

### A.5 PREDATOR-PREY CAPTURE RATE

Table 3 shows each method's success rate in capturing preys for Predator-Prey. CACL outperforms the baselines by capturing the most preys out of the evaluation episodes.

### A.6 POSITIVE LISTENING

This section describes the loss function we implemented for positive listening, based on Eccles et al. (2019). Given two policies $\pi^i$ and $\overline{\pi^i}$ of agent $i$ where the latter is the policy with messages zeroed out in the observations, and a trajectory $\tau$ of length $T$, the positive listening loss is written as:

| Learning Rate | 0.0003 |
|---|---|
| Epsilon for Adam Optimizer | 0.001 |
| $\gamma$ | 0.99 |
| Entropy Coefficient | 0.01 |
| Value Loss Coefficient | 0.5 |
| Gradient Clipping | 2500 |
| $\eta$ for *CACL* | 0.1 |
| $\kappa$ for *CACL* | 0.5 |
| Loss Coefficient for PL | 0.01 |
| Number of Asynchronous Processes | 12 |
| N-step Returns | 5 |

Table 4: Table for hyperparameters used across methods

$$L_{PL} = -\frac{1}{|T|} \sum_{j}^{T} \left[ \sum_{a \in A^i} (|\pi^i(a|\tau_j^i) - \overline{\pi^i}(a|\tau_j^i)|) + (\pi^i(a|\tau_j^i) \log(\overline{\pi^i}(a|\tau_j^i))) \right] \quad (6)$$

where in inner summation, the first term is the L1 Norm and the second term is the cross entropy loss.

### A.7 Message similarity for different goal locations in Find-Goal

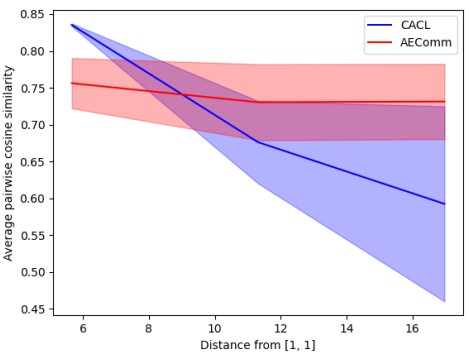

Figure 10: In Find-Goal, average pairwise cosine similarity between messages sent when the goal is at $[1, 1]$ and when the goal is in one of these coordinates $[5, 5], [9, 9], [13, 13]$. Similar to the representation probing test in section 5.6, we generate rollouts from the corresponding algorithms and filter for messages with the goal visible.

To further support that CACL can encode global information better, we measure the message similarity between messages sent when the goal is in $[1, 1]$ and when the goal is in other coordinates along the diagonal of the gridworld. As shown in Figure 10, CACL shows a clear downward trend in similarity as the distance from $[1, 1]$ increases unlike AEComm. This means CACL learns to encode global locations differently despite learning only with egocentric views. This cooroborates with the Goal Location Test in Table 2 that CACL is able to encode global information better.

### A.8 Protocol Representation Probing: 1-Layer

Table 5 shows the same results for the two probing tests in section 5.6 except here we use a 1-layer neural network instead of 2 layers. We observe significant dips in performance across all methods. Particularly, CACL becomes worse than the baselines in the easier Goal Visibility Test. However, CACL remains superior in the more difficult Goal Location test by an even bigger margin than the results in table 2.

Table 5: Classification results of the two probing tests in the Find-Goal environment, comparing all methods with communication. 1-layer neural networks are used for probing

|  | DIAL | PL | AEComm | CACL |
|---|---|---|---|---|
| Goal Visibility Test | $94.21\% \pm 2.68$ | $96.93\% \pm 3.14$ | $96.27\% \pm 3.98$ | $87.65\% \pm 3.86$ |
| Goal Location Test | $52.29\% \pm 5.25$ | $53.65\% \pm 9.60$ | $48.16\% \pm 7.34$ | $79.18\% \pm 5.63$ |

## A.9 ZERO-SHOT CROSS-PLAY

Table 6: Zero-shot cross-play performance in Predator-Prey. Best per-method results are bolded.

|  | CACL | AEComm | PL | DIAL |
|---|---|---|---|---|
| CACL (Ours) | $\mathbf{-17.20 \pm 5.14}$ | $\mathbf{-28.49 \pm 2.78}$ | $-24.61 \pm 5.77$ | $-28.78 \pm 3.99$ |
| AEComm |  | $-37.86 \pm 7.20$ | $-31.56 \pm 3.76$ | $-29.73 \pm 3.66$ |
| PL |  |  | $-27.07 \pm 2.94$ | $\mathbf{-22.89 \pm 3.98}$ |
| DIAL |  |  |  | $\mathbf{-22.85 \pm 2.04}$ |

Table 7: Zero-shot cross-play performance in Find-Goal. Best per-method results are bolded.

|  | CACL | AEComm | PL | DIAL |
|---|---|---|---|---|
| CACL (Ours) | $\mathbf{468.75 \pm 15.32}$ | $\mathbf{471.66 \pm 13.54}$ | $487.56 \pm 8.61$ | $488.28 \pm 16.60$ |
| AEComm |  | $479.96 \pm 14.96$ | $\mathbf{440.18 \pm 23.04}$ | $472.85 \pm 16.77$ |
| PL |  |  | $492.08 \pm 5.67$ | $486.41 \pm 10.46$ |
| DIAL |  |  |  | $\mathbf{476.07 \pm 15.89}$ |

An advanced form of coordination is working with partners you have not seen during training (Hu et al., 2020). Previous work has focused on coordination through actions (Carroll et al., 2019; Lupu et al., 2021) or pre-test grounding with a common dataset (Gupta et al., 2021) but to our knowledge, no previous work has succeeded in learning a linguistic communication protocol that is robust to zero-shot partners. To assess this advanced robustness, we take trained agents from different methods and random seeds and evaluate them with each other (i.e., zero-shot cross-play) in Predator-Prey and Find-Goal. Given two communication learning methods, $m_1$ and $m_2$, we sample two agents from each method for Predator-Prey and for Find-Goal, we average over sampling two agents from one method and one agent from the other and vice-versa. For intra-method cross-play, $m_1 = m_2$, we evaluate agents that were trained with the same method but from different random seeds, so they have not been trained with each other. For inter-method cross-play, $m_1 \neq m_2$, we sample agents from two different methods and pair them with each other. Each pairing is evaluated for 10 random seeds each with 10 evaluation episodes. Given that agents are trained in self-play (Tesauro, 1994) without regard for cross-play, we expect severe performance dips.

We show mean and standard deviation across random seeds for Predator-Prey and Find-Goal in Tables 6 and 7, respectively. As expected, all pairings take a significant dip in performance when compared with the main results. Inter-method cross-play performance is particularly bad across all algorithms. However, notably, CACL outperforms other methods in intra-method cross-play, indicating that the protocols learned by CACL are generally more robust even across random seeds. In general, zero-shot linguistic communication is incredibly difficult and our results are far from optimal. Still, CACL shows promise and demonstrates that contrastive SSL methods can lead to better zero-shot communication and coordination.

## A.10 BROADER IMPACT

More multi-agent learning systems will be deployed in the real-world as further progress is made in fields of multi-agent learning like MARL. We expect communication to play an essential role in these systems given how real-world problems are inherently complex and partially observable in most cases. Our focus on the decentralized communication setting contributes to the capability of learning more effective and consistent communication protocols. Having more consistent protocols improve mutual intelligibility and pave the way to multi-agent systems in which agents can communicate with unseen agents or even humans.

On the other hand, increasing adoption of such multi-agent learning systems could exacerbate certain risks. For instance, this could increase unemployment in a significant scale if systems operated by multiple humans like warehouses are replaced with multi-robot learning systems. It could also contribute to more advanced automated weaponry. In particular, given that our method explicitly considers messages sent from other agents in our protocol learning algorithm, it could encourage adversarial attacks which would lead to harmful behaviors and miscommunication, especially in mission-critical systems.

