# OpenReview forum: "Learning Multi-Agent Communication with Contrastive Learning"
_ICLR.cc/2024/Conference — ICLR 2024 poster_

### Official Review · Reviewer_Y1JN · 2023-10-22

**Soundness:** 2 fair
**Presentation:** 3 good
**Contribution:** 2 fair
**Rating:** 5
**Confidence:** 4

**Summary:**

This paper studies multi-agent communication by contrastive learning. It is motivated by the fact that communicated messages based on local observation can be viewed as incomplete views of the global state. From this perspective, a contrastive learning based approach is proposed, where states from close time steps are considered as positive samples, and those from distant time steps or episodes are considered as negative samples. The proposed algorithm is tested on several multi-agent benchmarks.

**Strengths:**

- The paper has an innovative perspective on the multi-agent communications, which motivates the use of contrastive learning.
- The paper is well-written, and the proposed contrastive learning framework is easy and clean to implement.
- There are various ablation studies over different components of the proposed algorithm and visualizations over the learned communicated messages.

**Weaknesses:**

- The improvement over the baselines is not obvious, especially in Traffic Junction in Figure 2. The standard error is also too large with a lot of overlaps, so it may need more seeds of experiments.
- The proposed algorithm CACL is only tested on three tasks. The paper could benefit from additional experiments on some more challenging tasks with partial observability where communications are intuitively beneficial.

**Questions:**

- The policy $\pi$ defined in section 3 seems to be only conditional on the local observation $\tau^i$. Should it also be conditional on the communicated messages?
- If some local observations miss important information, the approximated global states reconstructed by the message encoders may be very different from the true ones. Will this make the contrastive learning not meaningful?
- Does CACL require all-to-all communications, i.e., each agent communicate to all the other agents? If so, CACL is not scalable with large number of agents.
- During a time step, each agent receives multiple approximations of the global state from the communicated messages. This seems to include redundant information if the messages are not selectively received. How does CACL handle redundant information?

---

> ### Author Response · Authors · 2023-11-14
> **Rebuttal**
>
> Thank you very much for your helpful comments. We are glad that you find our work ‘innovative’ and ‘well-written. We hope we’ve addressed your concerns and questions below. Does the reviewer have any remaining concerns that prevent them from raising their score? We are happy to address them
>
> **More seeds**
>
> For fairness, we chose the number of seeds following our closest baseline AEComm. Note that CACL outperforms AEComm on Traffic Junction more than AEComm outperformed their own baseline in their paper. The task demonstrates that even if all methods fully solve a task, CACL can reach perfect reward faster.
>
> **More challenging environments**
>
> We use the same environments as baselines but made harder. E.g. in Predator-Prey, predators must fully surround a moving prey without being able to see each other, so communication is essential. In Find-Goal, we also reduce field-of-view to make communication key to good performance.
>
> **Policy is conditioned on messages**
>
> We include messages as part of the observation and have clarified this in section 3. Thank you for pointing it out.
>
> **CACL with different local information**
>
> CACL does not explicitly reconstruct the whole state with each message, but brings messages closer together. So a local observation without the goal cannot communicate goal location. But its message can communicate that the goal is *not* in that location, which should be meaningfully cohesive with another agents’ message containing goal location.
>
> **All-to-all communication is not necessary**
>
> Other agents’ messages act as positive / negative samples, so reducing all-to-all communication is possible, it just reduces the number of positive / negative samples. Scaling to more agents increases the number of samples so limiting communication directions should be completely fine.
>
> **Redundant information**
>
> You are correct that messages may contain redundant information, especially if the observations overlap. In our particular setup, this doesn’t seem to be a problem as the message encoder learns to extract useful information.

---

> > ### Author Response · Authors · 2023-11-17
> >
> > As the rebuttal period is coming to an end, we would like to thank you again for your helpful comments on our work. We hope that our clarifications, together with the additions to the revised manuscript, have addressed your concerns. Assuming this is the case, we would like to ask if you would be willing to update your review score. Otherwise, please let us know if you have any further questions. Thanks a lot.

---

> ### Comment · Reviewer_Y1JN · 2023-11-18
> **Thanks for your response**
>
> Thanks for the clarifications. Some of my concerns are addressed. However, I still have some reservations.
>
> - I don't find the results in figure 2 to be convincing due to very high standard errors. The curves are largely overlapped and can hardly be separated. More seeds are needed regardless of the number of seeds used in other works. The performance boosting of CACL over the baseline ACEComm is limited in Traffic-Junction.
>
> - Even if the selected baselines are made harder, I still think it is necessary for CACL to be tested on some commonly used MARL benchmarks with more complex tasks such as [1].
>
> - CACL has only been tested on scenarios with relatively small number of agents (<=5), so the redundancy/scaling issue may not be a problem. It is is not clear to me if this is still the case in scenarios such as 27m_vs_30m in [1] with 27 controllable agents.
>
> [1] StarCraft Multi-Agent Challenge

---

> > ### Author Response · Authors · 2023-11-20
> > **Response**
> >
> > Thank you for your quick reply. We understand your reservations and hope our responses shed light on our perspective.
> >
> > **Standard errors**
> >
> > We will run more seeds but may not have results in time for the end of discussion. We expect close results on Traffic Junction because we finely tuned our baselines, they are notably better than previous work [(Singh et al, 2018)](http://arxiv.org/abs/1812.09755). Our results on PP and FindGoal are still better despite strong baselines as well.
> >
> > **Starcraft**
> >
> > Very few works in MARL communication have evaluated on SMAC, and none use decentralized training. Just recently [SMACv2](https://openreview.net/pdf?id=5OjLGiJW3u) argues that SMAC is solvable without communication since its "partial observability is not particularly meaningful". We evaluate CACL in line with previous work. Compared to decentralized training baselines, we evaluate on harder, larger tasks ([AEComm](https://arxiv.org/abs/2110.15349)) and more tasks ([PL](https://arxiv.org/abs/1912.05676)).
> >
> > **Scaling**
> >
> > We believe the novelty of our method and connection to contrastive learning is our main contribution. Scaling is interesting and our method could be augmented with targeted or limited communication, as you suggested, to counter this issue. We believe our contribution is sufficient and hope to tackle scaling in future work.

---

> > > ### Author Response · Authors · 2023-11-23
> > >
> > > Thank you for the active discussion, we will include all suggestions in the final version of the paper. We appreciate the feedback and hope we've addressed the reviewer's main concerns to updating their score.

---

### Official Review · Reviewer_aKdr · 2023-10-27

**Soundness:** 4 excellent
**Presentation:** 4 excellent
**Contribution:** 3 good
**Rating:** 8
**Confidence:** 4

**Summary:**

The authors describe CACL, a contrastive learning approach for inducing communication among multiple agents. There are close parallels to classic contrastive learning methods in vision, for example, but the authors apply their technique to "emergent communication" to allow teams of agents to communicate with each other. "Postive" examples are grouped based on a window of recent timesteps, and, as in standard constrastive learning, agents learn to encode positive examples near each other.

In experiments, the authors show that CACL outperforms numerous baselines, including the SOTA AEComm method (which in some ways is similar in that it is a non-reward-based mechansim for inducing emergent communication).

**Strengths:**

I like this paper. It presents a simple idea that works well.

## Originality
Applying contastive losses to emergent communication is somewhat novel. (I know other works have also come out in this area, but they remain different in some important ways).

## Quality
The work is well-scoped and presented, with good results backing up claims.

## Clarity
I find the paper quite clear. Some figures could likely be redone to present the same information better (e.g., Figure 3), but mostly these are small changes.

## Significance
I think this work, should it be published, would be an important baseline for future emergent communication work.

**Weaknesses:**

Overall, this is a strong paper. To further improve the paper the authors could

1) Conduct further experiments to fill in Figure 4 in more detail (instead of just 3 or 4 checkpoints along the curve)

2) Run more trials, especially in the traffic junction where variance is high and not all methods seem to have converged.

**Questions:**

I have no outstanding questions. Overall, this was a clear paper.

---

> ### Author Response · Authors · 2023-11-14
> **Rebuttal**
>
> Thank you very much for your review. We are glad that you enjoyed our work and find it “clear”, “well-scoped [...] with good results”, and a “strong paper” that will be an “important baseline for future emergent communication work”. We are currently running more experiments to fill in Figure 4 as well as more trials for Traffic Junction, that will be included in the final paper. We would also be happy to rework Figure 3 to better present the information if the reviewer has suggestions.

---

> > ### Author Response · Authors · 2023-11-17
> >
> > As the rebuttal period is coming to an end, we would like to thank you again for your encouraging and helpful review on our work. We hope that our clarifications, together with the additions to the revised manuscript, have addressed your concerns and reinforced your positive opinion towards acceptance of our work.  Please let us know if you have any further questions. We would be happy to address them. Thanks a lot.

---

### Official Review · Reviewer_64q7 · 2023-10-30

**Soundness:** 3 good
**Presentation:** 3 good
**Contribution:** 2 fair
**Rating:** 6
**Confidence:** 4

**Summary:**

This paper proposes a new method for fully independent communication in MARL, CACL. The proposed method leverages the power of contrastive learning to ground communication and learning efficient communication for MARL tasks.

**Strengths:**

This paper tackles the problem of communication for fully independent learners, which is a very important topic in MARL and it is often underexplored. Also, mixing contrastive learning with MARL is interesting. Generally, the paper is well organised and well written.

**Weaknesses:**

Overall, this paper is interesting and investigates an important topic in MARL. However, I still have some concerns and questions that I would like the authors to comment on. Please find my comments below and questions ahead.

* The example of predator prey in figure 1 (right) is a bit confusing. I would not agree that the given examples correspond to similar views; for example, the first view (counting from the top) seems more similar to the third view rather than to the second view.
* In section 1, the authors mention: "we propose that an agent’s observation is a “view” of the environment state. Thus, different agents’ messages are encodings of different incomplete “views” of the same underlying state.". This is in fact the premise of a dec-pomdp; Observations are tipically local perceptions of the environment's state; in other communication methods in MARL where observation encodings are used as messages, I would say that the same logic is followed: individual perceptions of the environment are being shared as message encodings to the others. As described by the authors is section 3, the observations come from a function of the state.
* In the loss function, the RL loss is not defined. It would be good to have it for clarity purposes.
* A more detailed diagram of the architecture of this method could be beneficial to get a better understanding of the approach; the one presented in figure 8 in the appendix seems very simple and lacks detail and we cannot clearly understand how the gradients flow; this can be important since the authors are dealing with fully independent learning, without sharing parameters.

Minor:
* In section 3, do the authors mean $m_{t-1}^{-i}$ instead of $m_{t-1}^{-1}$?
* In section 3 "At each time step, each agent $i \in N$ chooses an action $a ∈ A^i$": shoud be $a^i$ since ahead $a$ is defined as the joint action.

**Questions:**

* I have questions about whether it is reasonable to evaluate the similarity of messages of different agents by simply looking at a window of a few timesteps in the trajectory. The observations corresponding to the generated messages can be different in important aspects from one timestep to the other, and thus would require distinct messages that could be biased by the contrastive loss. I am unsure whether this would scale to more complex cases, since it could not capture these differences in the observations.
* The experimented environments seem a bit simple and model scenarios where the observations can indeed be more similar to each other in some cases. Have the authors tested in more complex scenarios where there can be stronger variations on the observations such as SMAC? It would be interesting to see the performance in such complex environments.
* The authors mention that the setting followed is a fully decentralised setting where the agents do not share parameters or gradients (section 1); does this apply to the message encoder? I.e., do the agents share the same message encoder or does each one of them use a separate encoder to generate messages?
* I believe another potential direction for further work would be to investigate how to make methods such as the proposed one work together with the reward. I.e., in section 5.5 it was shown that currently it is detrimental. Yet, have the authors thought whether the method can be improved in any way in order to take advantage of the reward as well?

---

> ### Author Response · Authors · 2023-11-14
> **Rebuttal**
>
> We thank the reviewer for their detailed comments, corrections, and helpful suggestions and are glad they find our work interesting! We have implemented all the suggestions in the updated paper and answered all questions below. Are there any particular details that the reviewer would like cleared up in order to increase their score? We are happy to oblige.
>
> **Sliding Window Reasonability**
>
> Contrastive learning doesn’t force messages within a timestep to be exactly the same, just more similar, allowing for variation within the window. We expect it to work with more complex observations e.g. [TACO (Zheng et al, 2023)](https://arxiv.org/abs/2306.13229) use a similar time window for contrastive learning in single agent RL on DM control suite and Meta-World.
>
> **Environments with variable observations**
>
> All methods we have seen on SMAC use centralized training, so we choose environments with well known decentralized MARL + communication baselines, and make them harder. We expect more complex observations to make it easier to communicate some generic state info (e.g. location using visual cues) but make SSL harder, though contrastive learning should even more outperform autoencoding.
>
> **Message Encoder are not shared**
>
> Agents are fully decentralized and have completely separate message encoders, highlighting the difficulty of the task.
>
> **Figure 1 reworked**
>
> We have reworded the caption. Messages are positive samples if they are derived from [observations of] the same state. We’ve also changed the images to clarify our connection to image multi-view learning
>
> **Learning Representation with Reward**
>
> Our negative results reinforce a similar finding in AEComm, and likely stem from the two training signals (SSL and RL) conflicting, causing instability. A possible future approach could alternate training one signal then the other or add an attention layer that conditions on the state to decide on a weighted mixture of the two losses.
>
> **CACL vs Dec-POMDP**
>
> This is exactly our view. CACL’s innovation is to leverage the premise of Dec-POMDP to explicitly capture state info in the message. Other methods (e.g. AEComm) reconstruct observation from message, missing key state info.
>
> **RL Loss Added**
>
> Thank you for pointing that out. We have added a RL loss in Appendix A.1 for clarity, with text in red.
>
> **Figure 8 reworked**
>
> Thank you for the suggestion. Some details were in Appendix A.2, we’ve added them to the figure and illustrate gradient flow of each loss, with caption text in red.
>
> **Section 3 notation**
>
> Thank you for finding these minor errors. We have corrected it to m_{t-1}^{-i} and a^i  in the updated paper in red.

---

> ### Author Response · Authors · 2023-11-17
>
> As the rebuttal period is coming to an end, we would like to thank you again for your detailed comments and helpful suggestions on our work. We hope that our clarifications, together with the additions to the revised manuscript, have addressed your concerns. Please let us know if you have any further questions we can address in order for the reviewer to increase their score.

---

> > ### Author Response · Authors · 2023-11-21
> >
> > As the rebuttal period is ending imminently and we have not heard back from the reviewer regarding our rebuttal, we would like to reach out again and see if there are any outstanding questions and concerns. Please let us know if you have any further questions we can address in order for the reviewer to increase their score.

---

> > > ### Comment · Reviewer_64q7 · 2023-11-22
> > > **Thanks for the response**
> > >
> > > Thank you for your response. After reading the comments, most of my concerns were addressed. While I am aware that usually approaches in SMAC follow centralised training, that does not mean that decentralised approaches should not be able to solve these environments. In fact, I think it would demonstrate improved robustness to have a decentralised method capable of solving the complex communication environments presented by the authors, but that can also solve other complex tasks where usually centralisation is assumed, such as SMAC.
> > >
> > > Overall, the method to ground communication is interesting and I believe that decentralised control is important. Thus, I will raise my score.

---

> > > > ### Author Response · Authors · 2023-11-22
> > > >
> > > > Thank you for your response and for raising the score. We are glad that most of your concerns have been addressed. We fully agree that it is essential to scale decentralised methods to more complex setting to fully realise the benefits of decentralisation. We hope to further scale decentralised communications methods to larger-scale MARL environments in the future. Thanks again.

---

### Official Review · Reviewer_sueF · 2023-11-01

**Soundness:** 1 poor
**Presentation:** 2 fair
**Contribution:** 2 fair
**Rating:** 5
**Confidence:** 4

**Summary:**

This paper presents a novel approach to guide multi-agent communication learning via contrastive learning in a decentralized MARL scenario. The intuition is that under similar circumstances the agents should emit similar messages, and vice versa. Hence the authors employ contrastive learning to maximize the mutual information between messages of a given trajectory and minimize other cases. The authors claimed their method has outperformed exisiting approaches on several benchmarks.

**Strengths:**

The idea delivered by this work is clear and somewhat grounded. Indeed it would be worthwhile for agent to learn a guidance of its message during multi-agent communication. And the intuition of enforcing messages under similiar state to be alike with each other is a straightforward motivation, for which contrastive learning might be one of the most popular method to achieve.

**Weaknesses:**

However, after going through the whole paper, It is easy to find that the proposed idea is less sufficiently proved and there are many flaws in the manuscript. There are a few such perspectives:
1. In section 4, the negative samples are defined as from outside the current time window or other trajectories. This is not technically sound since it would be possible for agents to encounter similar states at different trajectories (which would be considered as negative by the proposal). It is suggested that the authors should discuss such cases in detail and figure out more solid principle to decide positive/negative samples for contrastive learning.
2. The selected benchmark for comparison is kind of limited. Public MARL evaluation platforms like SMAC[1] or MATE[2] which involves higher complexity should be considered for more pursuasive comparison. In addition, in the compared task of Traffic Junction, the improvement seems to be marginal.
3. The compared baseline methods are sort of obselete. Newer published works in recent 3 years should get included. (especially new work in 2022-2023).

[1] Samvelyan, M., Rashid, T., De Witt, C. S., Farquhar, G., Nardelli, N., Rudner, T. G., ... & Whiteson, S. (2019). The starcraft multi-agent challenge. arXiv preprint arXiv:1902.04043.
[2] Pan, X., Liu, M., Zhong, F., Yang, Y., Zhu, S. C., & Wang, Y. (2022). Mate: Benchmarking multi-agent reinforcement learning in distributed target coverage control. Advances in Neural Information Processing Systems, 35, 27862-27879.

**Questions:**

More comprehensive comparison and analysis are expected:
1. It is encouraged for the authors to demonstrate the scalability of the proposed approach, like in a continuous state/action space environments which may involve a large number of agents with quite dense communication load. In such cases, would the proposed scheme be better than the most recent communication-based work like ATOC[1], MF-MARL[2], TarMAC[3], I2C[4], ToM2C[5]?
2. Besides showing the similarity of messages among multiple states, the exact improvement from such a contrastive learning method should be analyzed. For instance, it is better to compare the adjacency/disparity of positive/negative sample pairs before/after the proposed training.

[1] Jiang, J., & Lu, Z. (2018). Learning attentional communication for multi-agent cooperation. Advances in neural information processing systems, 31.
[2] Yang, Y., Luo, R., Li, M., Zhou, M., Zhang, W., & Wang, J. (2018, July). Mean field multi-agent reinforcement learning. In International conference on machine learning (pp. 5571-5580). PMLR.
[3] Das, A., Gervet, T., Romoff, J., Batra, D., Parikh, D., Rabbat, M., & Pineau, J. (2019, May). Tarmac: Targeted multi-agent communication. In International Conference on Machine Learning (pp. 1538-1546). PMLR.
[4] Ding, Z., Huang, T., & Lu, Z. (2020). Learning individually inferred communication for multi-agent cooperation. Advances in Neural Information Processing Systems, 33, 22069-22079.
[5] Wang, Y., Zhong, F., Xu, J., & Wang, Y. (2021). Tom2c: Target-oriented multi-agent communication and cooperation with theory of mind. arXiv preprint arXiv:2111.09189.

---

> ### Author Response · Authors · 2023-11-14
> **Rebuttal**
>
> Thank you for your helpful feedback, we hope we have addressed your comments below. Given the borderline score, are there any other concerns or questions the reviewer would like addressed in order to increase their score?
>
> **CACL’s negative sample window is sufficient**
>
> Although edge-cases can exist, contrastive learning does not require perfect negative mining, just a good general rule. E.g. [TACO (Zheng et al, 2023)](https://arxiv.org/abs/2306.13229) uses a similar time window for negative samples in contrastive learning for single agent RL.
>
> **Our benchmarks are thorough and robust**
>
> We extend and harden benchmarks used in baselines, e.g. AEComm, see section 5.1 for details, and they are nearly identical or harder versions of benchmarks in the most recent papers you reference (I2C and ToM2C). SMAC does not generally require communication. MATE is interesting but all of its communication baselines use centralized training and we focus on benchmarks with existing decentralized training baselines.
>
> **Baselines are SOTA**
>
> Our baselines are current state-of-the-art in decentralized communication. Can the reviewer point to other decentralized RL + communication methods? All noted papers seem to be centralized training methods.
>
> **Evaluating Contrastive Learning**
>
> As requested, we plot the distance between positive and negative samples (i.e. our contrastive loss) over the course of training in Figure 9 in Appendix A.4. As training evolves, contrastive loss decreases so positive and negative samples become more separated.
>
> **Scalability of CACL**
>
> All noted works (ATOC, TarMAC,...) are centralized training methods whereas CACL is decentralized training which should be better at scaling to large number of agents [(Lyu et al, 2021 Section 6.3)](https://arxiv.org/pdf/2102.04402.pdf). We do not investigate communication overhead, but targeting methods (e.g. TarMAC) could be applied to CACL as well.

---

> > ### Author Response · Authors · 2023-11-17
> >
> > As the rebuttal period is coming to an end, we would like to thank you again for your valuable feedback on our work. We hope that our clarifications, together with the additions to the revised manuscript,  have addressed your concerns. Assuming this is the case, we would like to ask if you would be willing to update your review score. Otherwise, please let us know if you have any further questions. Thanks a lot.

---

> > ### Comment · Reviewer_sueF · 2023-11-18
> > **Reply to rebuttal**
> >
> > As claimed in Sec.1 of the manuscript, "Centralized training with decentralized execution (CTDE) (Lowe et al., 2017) is a middle-ground between purely centralized and decentralized methods but may not perform better than purely decentralized training (Lyu et al., 2021). " It would be better for the author to comprehensively prove such argument with firm results, otherwise it is unclear how the proposed approach compare with SOTA effort on CTDE setting. Especially when current transformer based centralized training can afford a higher complexity for multi-agent communication compared with previous RNN/LSTM modules, it would be uncertain whether a decentralized training is really necessary. Since CTDE has been a publicly approved setting for MARL, these works shouldn't be ignored and need appropriate citations & fair comparisons.

---

> ### Author Response · Authors · 2023-11-18
>
> Thank you for your active discussion. We agree that CTDE is an interesting setup but believe that decentralized training is a separate research area with important research questions that we tackle in our work.
>
> **Justifying Decentralized**
>
> We cite [Lyu et al. 2021](https://arxiv.org/pdf/2102.04402.pdf) because they show theoretically and empirically that decentralized critics can match or exceed CTDE performance on a variety of tasks and refer to their work for comprehensive proof. Empirically, decentralized seems to match CTDE performance e.g. CTDE MAPPO is similar to decentralized IPPO on many tasks including Starcraft (Table 4 in [Yu et al. 2022](https://arxiv.org/abs/2103.01955)). Decentralized training is desirable in real-world scenarios where you don't have synchronous information for all agents ([Li et al. 2023](https://jmlr.org/papers/volume24/20-700/20-700.pdf)) and it is generally a well-studied area of research (see related work in [Lu et al, 2022](https://arxiv.org/pdf/2211.03032v1.pdf) and section 4 in [Oroojlooy and Hajinezhad, 2021](https://arxiv.org/pdf/1908.03963.pdf)).
>
> **Why CACL is decentralized**
>
> Where CTDE's issue is scalability ([Lyu et al, 2021](https://arxiv.org/pdf/2102.04402.pdf)), decentralized training suffers in stability, especially for learning communication. CACL's contrastive learning provides an elegant framework to learn communication while explicitly solving the problem of consistency in decentralized protocols (which implicitly improves stability). Centralized controllers could easily enforce a uniform, symmetric communication but it is novel to achieve this with decentralized training.

---

### Author Response · Authors · 2023-11-15
**Updated Paper**

We would like to thank the reviewers for their time and insightful feedback. We are glad they find our work “grounded” (sueF), “an important topic” (64q7), “novel” / “innovative” (akDr, Y1JN) and “good results backing up claims” (akDr).  We have updated the manuscripts according to their suggestions, here is a summary of the main changes:
- Updated Figure 1 to clarify our connection to image multi-view learning, pointed out by Reviewer 64q7
- Added RL loss equations to Appendix A.1., pointed out by Reviewer 64q7
- Updated architecture figure to indicate gradient flow in Figure 8, Appendix A.3, pointed out by Reviewer 64q7
- Added a new plot of the contrastive loss over training in Figure 9 in Appendix A.4, to show how positive/negative samples become more separable, as requested by Reviewer sueF

---

### Meta-Review · Area_Chair_bK8x · 2023-12-19

**Metareview:**

The paper presents a novel approach to multi-agent communication learning using contrastive learning in decentralized MARL scenarios. The reviewers appreciated the interesting use of contrastive learning, the clarity and motivation of the paper. However, there were concerns about the technical soundness, limited benchmark selection, scalability, and unclear improvements over baselines. For these reasons, I recommend acceptance, but I encourage the authors to address these concerns in the final manuscript.

**Justification For Why Not Higher Score:**

Concerns about the technical soundness, limited benchmark selection, scalability, and unclear improvements over baselines.

**Justification For Why Not Lower Score:**

Interesting use of contrastive learning, the clarity and motivation of the paper.

---

### Decision · Program_Chairs · 2024-01-16

Accept (poster)